# Coverage of antenatal, intrapartum, and newborn care in 104 districts of Ethiopia: A before and after study four years after the launch of the national Community-Based Newborn Care programme

Della Berhanu[1,2]*, Elizabeth Allen[3], Emma Beaumont[3], Keith Tomlin[4], Nolawi Taddesse[5], Girmaye Dinsa[6,7], Yirgalem Mekonnen[5], Hanna Hailu[5], Manuela Balliet[8], Neil Lensink[9], Joanna Schellenberg[1], Bilal Iqbal Avan[10]

1 Department of Disease Control, London School of Hygiene and Tropical Medicine, London, United Kingdom, 2 Health System and Reproductive Health Research Directorate, Ethiopian Public Health Institute, Addis Ababa, Ethiopia, 3 Department of Medical Statistics, Faculty of Epidemiology and Population Health, London School of Hygiene and Tropical Medicine, London, United Kingdom, 4 Department of Population Health, Faculty of Epidemiology and Population Health, London School of Hygiene and Tropical Medicine, London, 5 JaRco Consulting, Addis Ababa, Ethiopia, 6 Harvard T.H. Chan School of Public Health, Harvard University, Boston, United States of America, 7 Department of Public Health and Health Policy, College of Health and Medical Sciences Haramaya University, Ethiopia, 8 Department of Global Health and Development, London School of Hygiene and Tropical Medicine, London, United Kingdom, 9 Global Affairs Canada, Ottawa, Canada, 10 Department of Clinical Research, Faculty of Infectious and Tropical Diseases, London School of Hygiene and Tropical Medicine, London, United Kingdom

* della.berhanu@lshtm.ac.uk

**Data Availability Statement:** Study datasets for the Community Base Newborn Care Evaluation (CBNC) are made available on LSHTM Data

## Abstract

### Background

Access to health services across the continuum of care improves maternal and newborn health outcomes. Ethiopia launched the Community-Based Newborn Care programme in 2013 to increase the coverage of antenatal care, institutional delivery, postnatal care and newborn care. The programme also introduced gentamicin and amoxicillin treatment by health extension workers for young infants with possible serious bacterial infection when referral was not possible. This study aimed to assess the extent to which the coverage of health services for mothers and their young infants increased after the initiation of the programme.

### Methods

A baseline survey was conducted in October-December 2013 and a follow-up survey four years later in November-December 2017. At baseline, 10,224 households and 1,016 women who had a live birth in the 3–15 months prior to the survey were included. In the follow-up survey, 10,270 households and 1,057 women with a recent live birth were included. Women were asked about their experience of care during pregnancy, delivery and postpartum periods, as well as the treatment provided for their child's illness in the first 59 days of life.

Compass (https://doi.org/10.17037/DATA.
00001979), a research data repository operated by
the London School of Hygiene and Tropical
Medicine. To protect participant confidentiality, the
datasets are made available through a controlled
access approach. Although CBNC datasets do not
contain direct identifiers (such as names),
repository staff indicate there is a risk that
participants may be indirectly identifiable in some
circumstances, due to the large number of
variables collected from each participant and the
accuracy of measures. Researchers wishing to
access datasets are asked to apply for access via
the repository's data request form, providing
information on the variables they wish to access
and details of their analysis plan. The request will
be sent to the study team (that performed the
research and have greatest understanding of the
data) and the LSHTM Research Data Manager
(who acts as an independent advisor). If the data
analysis can be performed in compliance with the
study's ethical and legal requirements, the study
team will produce a derived dataset that contains
the requested variables and work with the applicant
to help them to understand the data. If there
remains a recognisable risk that the derived dataset
contains potentially identifiable information, the
applicant will be asked to sign a Data Sharing
Agreement before being provided with the dataset.

**Funding:** This project was funded by Bill & Melinda
Gates Foundation (https://www.gatesfoundation.
org/) [OPP1017031]. JS received the award. The
funder had no role in the study design, data
collection and analysis, decision to publish, or
preparation of the manuscript.

**Competing interests:** The authors have declared
that no competing interests exist.

**Abbreviations:** ANC, antenatal care; CBNC,
community-based newborn care; HEW, health
extension worker; PNC, postnatal care; UNICEF,
United Nation Children's Fund.

## Results

Between baseline and follow-up surveys the proportion of women reporting at least one antenatal care visit increased by 15 percentage points (95% CI: 10,19), four or more antenatal care visits increased by 17 percentage points (95%CI: 13,22), and institutional delivery increased by 40 percentage points (95% CI: 35,44). In contrast, the proportion of newborns with a postnatal care visit within 48 hours of birth decreased by 6 percentage points (95% CI: -10, -3) for home deliveries and by 14 percentage points (95% CI: -21, -7) for facility deliveries. The proportion of mothers reporting that their young infant with possible serious bacterial infection received amoxicillin for seven days increased by 50 percentage points (95% CI: 37,62) and gentamicin for seven days increased by 15 percentage points (95% CI: 5,25). Concurrent use of both antibiotics increased by 12 percentage points (95% CI: 4,19).

## Conclusion

The Community-Based Newborn Care programme was an ambitious initiative to enhance the access to services for pregnant women and newborns. Major improvements were seen for the number of antenatal care visits and institutional delivery, while postnatal care remained alarmingly low. Antibiotic treatment for young infants with possible serious bacterial infection increased, although most treatment did not follow national guidelines. Improving postnatal care coverage and using a simplified antibiotic regimen following recent World Health Organization guidelines could address gaps in the care provided for sick young infants.

## Introduction

Ethiopia achieved the Millennium Development Goal of reducing under-5 mortality by two-thirds, two years ahead of the 2015 target [1, 2]. The success was attributed to a multisectoral approach taken by the country, including an expansion in the primary health care services and progress made in improving child nutrition [2]. Despite the success in averting under-5 deaths, the reduction in neonatal mortality (deaths in the first 28 days of life) was marginal. Between 2005 and 2011 neonatal mortality decreased from 39 to 37 deaths per 1000 live births, and by 2013 neonatal mortality accounted for 43% of under-5 deaths [1, 3, 4].

The low coverage of antenatal care (ANC), institutional delivery and postnatal care (PNC) all contribute to high neonatal mortality in Ethiopia [5]. The 2014 Mini Ethiopian Demographic and Health Survey showed that the coverage of four or more ANC visits was 32%, and just 15% and 12% for facility delivery and a PNC care visit in the first 48 hours, indicating inadequate care seeking for mothers and newborns [6]. Cultural practices, financial constraints, the perceived quality of care provided by community health workers, lack of caregivers' knowledge of danger signs, distance to health posts, health post closure and stock out of drugs are all reported as barriers to care seeking [7–12].

In March 2013 the government of Ethiopia launched the Community-Based Newborn Care (CBNC) programme, which had nine components aiming to provide lifesaving services for mothers and newborns [2, 13]. The strategies included early identification of pregnant women, provision of focused ANC, promotion of facility delivery and management of sick newborns [14, 15]. The CBNC programme was integrated into the existing health extension programme which is the platform for primary health care service delivery. The health extension programme, which was launched in 2003 to bring health services closer to rural

communities, is structured around the primary health care unit, comprising five health posts, their referral health centre and a primary hospital [16, 17]. Comprehensive community-based maternal, newborn and child health services are a component of the health extension programme. The services are delivered by female community health workers known as health extension workers (HEWs), who are based at health posts. In theory, two HEWs provide services for a population of 5000 people, both through outreach activities and static services at health posts. They are assisted by a volunteer women's group known as the Women's Development Army leaders, which was established in 2011 [18].

In 2013, preterm birth complications accounted for 35% of global neonatal deaths, with 24% due to complications during labour and delivery, and 15% due to sepsis [19]. Evidence also suggests that prematurity, asphyxia and infections account for most neonatal deaths in Ethiopia [20, 21]. The CBNC programme aimed to reduce these causes of neonatal mortality, and particularly those from infections, through the "four Cs": early **C**ontact with prenatal and postnatal women, **C**ase-identification of newborns with possible infection, **C**are or treatment of sick newborns, and **C**ompletion of appropriate antibiotics for seven days [22]. As such the task of treating young infants (0–59 days) with possible serious bacterial infection was shifted from health care providers at health centres or hospitals to HEWs when referral was not possible. HEWs were trained to provide injectable and oral antibiotics [13], in the light of global evidence that community health care workers, if trained and supported, could identify and treat newborns with possible serious bacterial infections [23]. The CBNC programme also built on the lessons learned from the integrated Community Case Management of common childhood illnesses programme initiated in 2010 and the Community-Based Interventions for Newborns in Ethiopia trial, which was an ongoing cluster-randomized study at the time the CBNC programme was launched. Both indicated that HEWs could be trained to identify and treat newborns with possible serious bacterial infection [24, 25].

This study aimed to assess the extent to which the coverage of services along the continuum of care increased after the launch of the CBNC programme, by comparing household-level changes over time for women who had delivered in the previous 3–15 months.

## Materials and methods

### Description of the Community-Based Newborn Care programme

The CBNC programme had nine components along the continuum of care (Fig 1): (1) early identification of pregnancy, (2) provision of focused ANC, (3) promotion of institutional delivery, (4) safe and clean delivery, (5) immediate newborn care including application of chlorhexidine on cord, (6) recognition of asphyxia, initial stimulation and resuscitation, (7) prevention and management of hypothermia, (8) management of pre-term and low birth weight neonates, and (9) management of possible serious bacterial infection in young infants (0–59 days) at the community level when referral was not possible. A detailed description of the CBNC programme intervention is provide in S1 Table.

Although CBNC programme aimed to promote facility delivery, components 4–6 attempted to address the care HEWs could provide to newborns that are delivered at home or in health posts. Furthermore, of the nine CBNC programme components, seven of them were part of the existing services, while use of chlorhexidine on the cord and the treatment of possible serious bacterial infection at the community level when referral was not possible were new components. If referral was possible, HEWs were trained to provide a pre-referral dose of gentamicin and amoxicillin. A young infant with any one of the following symptoms was considered to have possible serious bacterial infection: convulsions, reduced or no feeding, high (>37.5˚C) or low (<35.5˚C) temperature, fast breathing, no or limited movement and severe

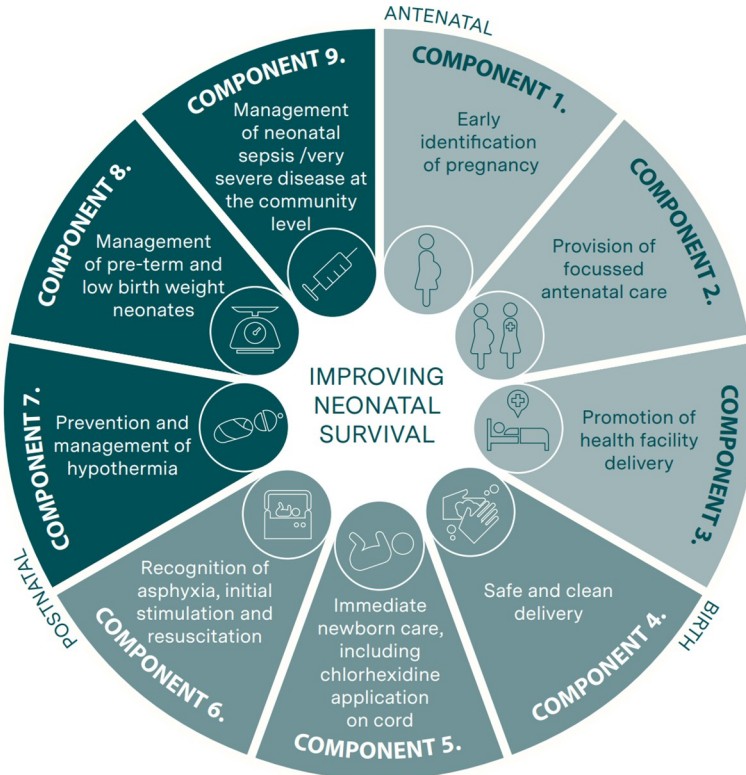

**Fig 1. The nine components of Community-Based Newborn Care programme.**

chest in-drawing. Prior to the implementation of the CBNC programme, the management guidelines for young infants with any of these danger signs was immediate referral to a health centre where they would be treated with gentamicin and ampicillin.

## Rollout of the Community-Based Newborn Care programme

The CBNC programme was led by the Federal Ministry of Health and implemented with support from United Nations Children's Fund (UNICEF), Save the Children, Last 10 Kilometres/John Snow Inc., and the Integrated Family Health Programme. It was rolled out in three phases in four regions of Ethiopia: Amhara, Tigray, Oromia, and Southern Nations, Nationalities and Peoples (Fig 2). These four regions had approximately 570 districts. In March 2014, all HEWs in 104 of these districts were trained in the nine components of the CBNC programme and the content of their training is provided is S2 Table. These districts were selected by the Federal Ministry of Health for: reporting high utilization for integrated Community Case Management services; having a strong linkages within their primary health care units; having a well-established health extension programme; and, having functional Women's Development Army networks. In January 2015, a second phase of training was initiated for all HEWs in the remaining districts located in the four regions. As part of phase 3, in 2018, the CBNC programme was rolled out in the remaining regions of the country.

## Study setting and design

Ethiopia is organized by region, zone, woreda (district), kebele (village) and gote (sub-village). This study was conducted in 52 of the phase 1 districts located across 7 zones and 49 of the

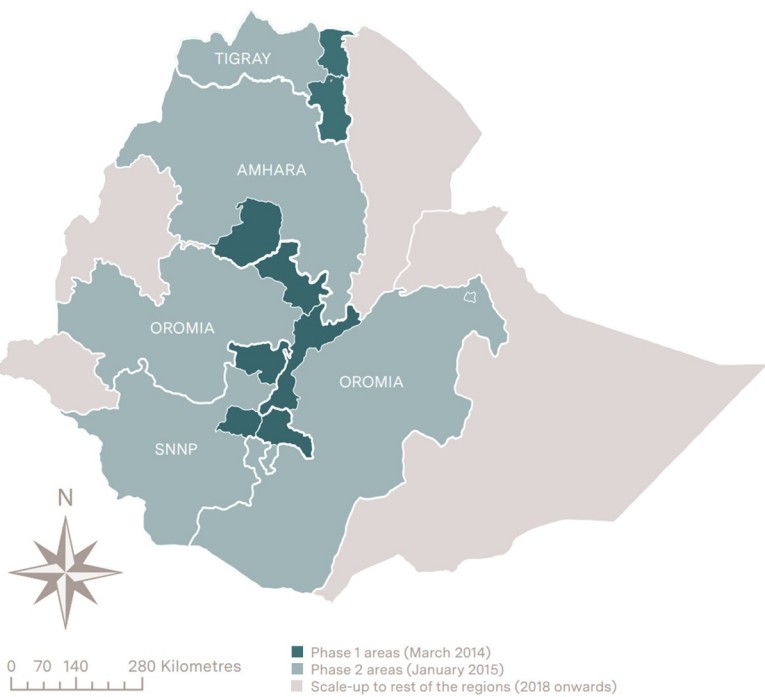

**Fig 2. The three phases of Community-Based Newborn Care programme implementation.** Produced by the authors using ArcGIS.

phase 2 districts located in 5 zones (Fig 3). We used a before and after population-based cross-sectional survey of the CBNC programme service coverage at the household level. The baseline survey was conducted in October-December 2013 and the follow-up survey four years later in November-December 2017. Fig 4 shows the timing of the surveys along with the timing of CBNC programme implementation. The evaluation was conducted by the London School of Hygiene and Tropical Medicine, in collaboration with JaRco Consulting.

## Sample size

The 2011 Ethiopian Demographic and Health Survey fertility rate of 4.8 births per woman indicated that a cross-sectional survey would yield on average one woman aged 13–49 years with a birth in the 3 to 15 months prior to the survey in 10% of households. As such, 10,450 households would allow us to detect a difference in coverage rates between intervention and comparison areas of at least 10 percentage points in key indicators including antenatal care, institutional delivery and post-natal care using baseline coverage rates from the 2011 Ethiopian Demographic and Health survey, with 80% power, 5% significance, a design effect of 1.4 and 90% completeness [4]. Similarly, an assumption that 5–10% of newborns would have possible serious bacterial infection within the first 30 days after birth was expected to yield 50 to 100 cases in this sample of households. If 10% of newborns received treatment for possible serious bacterial infection at baseline, a sample of 100 cases would allow us to detect a 23-percentage point difference between intervention and comparison areas with 80% power and 5% level of significance. No design effect or attrition was allowed for this stage. The above sample size was calculated to measure a difference in CBNC service coverage after four years of the CBNC programme implementation using a quasi-experimental design comparing phase 1 and phase 2 areas. However, implementation in phase 2 areas went ahead sooner than expected, and our study design

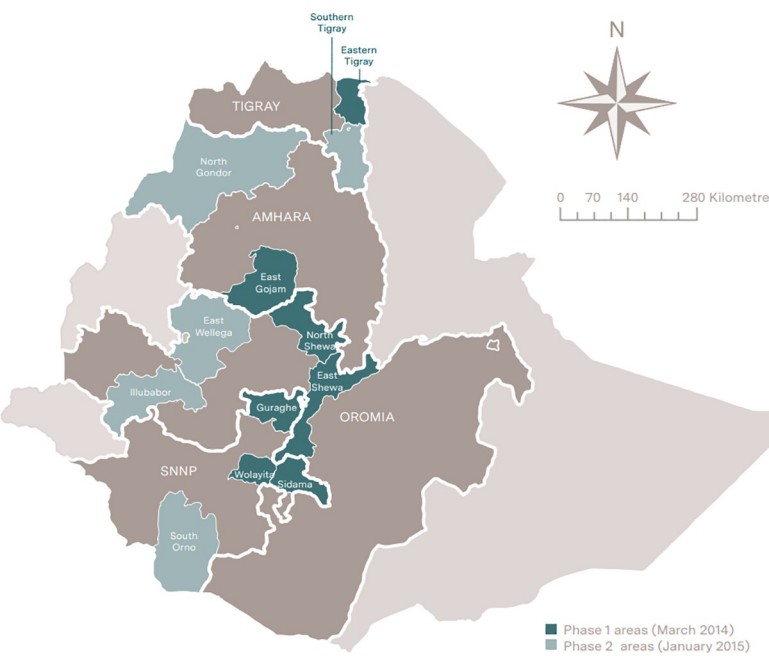

**Fig 3. Community-Based Newborn Care evaluation zones.** Produced by the authors using ArcGIS.

became a less robust before-and-after comparison, but with roughly double the sample size originally planned. Moreover, although our sample size estimates for treatment of sick young infants was based on the first month of life, the programme defined sick young infants as those under 60 days old, and we recruited and analysed infants aged 0–59 days in our study.

## Sampling procedure and study participants

For the baseline (2013) survey we used simple random sampling to select half of the districts in each of the 7 phase 1 zones (52 out of 104 districts) and a similar number of districts (49 out of 70) in the five of the phase 2 zones. In all of the 52 phase 1 districts and 47 out of the 49 phase 2

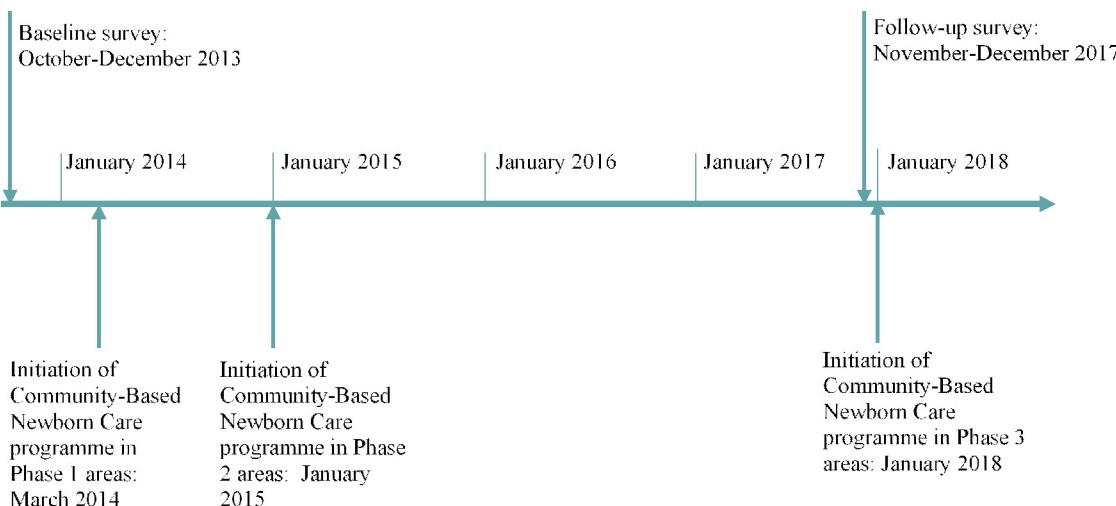

**Fig 4. Timeline for the Community-Based Newborn Care programme implementation and for the baseline and follow-up surveys.**

districts, we visited two primary health care units that were selected using simple random sampling. In two phase 2 districts, which were the only two agrarian districts in a predominantly pastoralist zone (South Omo), we included 11 primary health care units, to obtain similar total number of primary health care units as in phase 1 areas. Three primary health care units from phase 2 areas were excluded from the study at the time of the baseline survey due to security and accessibility issues resulting in a total of 206 primary health care units that were included in the study. They were also excluded from the follow-up survey. In each primary health care unit one kebele (village), and within the kebele one gote (sub-village) were selected for inclusion using simple random sampling. All households in the gote were listed and 50 were randomly chosen for the study. All the selected households were visited, and no replacement was made for households where no one was available after three visits or where consent was not provided. In each household we interviewed the household head to assess the demographics of the household, and all women between 13–49 years of age were asked about their birth history. Women who gave birth in the 3 to 15 months prior to the survey were asked additional questions about the experience of care during pregnancy, delivery and postpartum periods, as well as the treatment provided for their child if he/she was sick in the first 59 days of life. We visited the same kebeles for the follow-up survey and followed the same procedures to select households and study participants.

## Data collection and management

The survey tool included previously validated questions as well as questions that were specifically developed for the CBNC programme [26]. Extensive formative research and pretesting was conducted to develop the survey tool. Once finalised the tools were translated from English (S1 File) to Amharic (S2 File) and then back translated. The Amharic tool was then translated into Tigrinya (S3 File) and Oromifa (S4 File) and then back translated to Amharic. Data were collected through paper questionnaires at baseline and Samsung tablet model SM-T210 for the follow-up survey. Tablets were programmed using CSPro Version 7.1.1. We selected data collectors who had a minimum of bachelor's degree, previous experience with data collection and the ability to speak the required language. Training was provided over the course of six days and included a pilot test. The data collection teams each had one supervisor and four data collectors. Supervisors randomly accompanied data collectors and randomly re-interviewed 20% of study participants in each household cluster to ensure quality and consistency. They ensured that sampling procedures were followed and reviewed questionnaires for content and completeness. There were also regional coordinators that made field visits and contacted supervisors daily to address questions and concerns. For baseline survey, questionnaires were transported regularly from the field to JaRco Consulting's central office where 100% were double entered into CSPro and reconciled. For the follow-up survey data were uploaded to JaRco Consulting's central server daily or whenever internet was available. Data were coded and cleaned, checking for consistency and completeness. During the study data were stored on both JaRco Consulting's and the London School of Hygiene and Tropical Medicine's server and access was restricted to members of the research team only. On completion of the study, cleaned, labelled and fully anonymised data sets, and the questionnaire, were made available on the London School of Hygiene and Tropical Medicine's data repository, Data Compass (https://doi.org/10.17037/DATA.00001979).

## Outcome variables

For women who had a live birth in the 3 to 15 months prior to the day of the survey, we assessed indicators that capture the nine components of the CBNC programme as shown in Table 1. For clean and safe delivery, immediate newborn care, recognition and management of

**Table 1. Outcome indicators for the community-based newborn care programme identified for change between baseline and follow-up surveys.**

| CBNC programme components | Outcomes measured |
| --- | --- |
| 1. Early identification of pregnancy | % of women 13–49 years of age with a live birth in the 3–15 months prior to the survey who reported one or more antenatal care visit |
| | % of women 13–49 years of age with a live birth in the 3–15 months prior to the survey who reported informing a woman development army leader of their pregnancy |
| 2. Focused antenatal care | % of women 13–49 years of age with a live birth in the 3–15 months prior to the survey who reported four or more antenatal care visits |
| | Among women with one or more antenatal care visit, % who, at any of their visit, reported: being counselled on birth prerenders, pregnancy danger signs, breastfeeding and nutrition; having their weight, height and blood pressure measured; giving urine sample; and, receiving iron folate, syphilis testing and HIV testing |
| 3. Promotion of institutional delivery | % of women 13–49 years of age with a live birth in the 3–15 months prior to the survey who reported delivering in a health centre or hospital |
| 4. Clean and safe delivery:[a] | % of women 13–49 years of age with a live birth in the 3–15 months prior to the survey who reported that their birth assistant washed his or her hands with soap before assisting in the delivery |
| | % of women 13–49 years of age with a live birth in the 3–15 months prior to the survey who reported that their birth assistant wore gloves during the delivery |
| | % of women 13–49 years of age with a live birth in the 3–15 months prior to the survey who reported that they had delivered on a clean surface |
| 5. Immediate newborn care:[a] | % of children born in the 3–15 months prior to the survey where a clamp or new or boiled thread or string was used to tie their cord, as reported by their mothers |
| | % of children born in the 3–15 months prior to the survey where a new razor blade or sterilized scissors was used to cut their cord, as reported by their mothers |
| | % of children born in the 3–15 months prior to the survey where antiseptic was applied on their cord, as reported by their mothers |
| | % of children born in the 3–15 months prior to the survey who received a postnatal care visit in the first 48 hours after birth, as reported by their mothers |
| 6. Recognition and management of asphyxia:[a] | % of children born in in the 3–15 months prior to the survey who had difficulty breathing at birth, as reported by their mothers |
| | Among children born in in the 3–15 months prior to the survey who had difficulty breathing at birth, % who were resuscitated, as reported by their mothers |
| 7. Prevention and management of hypothermia:[a] | % of children born in the 3–15 months prior to the survey who were placed on their mother's belly or chest immediately after birth, as reported by their mothers |
| | % of children born in the 3–15 months prior to the survey whose bathing was delaying for 24 hours after birth, as reported by their mothers |
| 8. Management of pre-term and low birth weight neonates:[a] | % of children born in the 3–15 months prior to the survey who were weighed at birth, as reported by their mothers |
| 9. Management of possible serious bacterial infection in young infants:[a] | % of children born in the 3–15 months prior to the survey who had signs of possible serious bacterial infection in the first 59 days of life, as reported by their mothers |
| | Among children born in the 3–15 months prior to the survey who had signs of possible serious bacterial infection in the first 59 days of life, % who received: gentamicin for seven days; amoxicillin for seven days; and, gentamicin and amoxicillin for seven days. |

[a] These indicators were calculated separately for institutional and home deliveries.

[b] Signs of possible serious bacterial infection: unusually hot or unusually cold, reduced feeding, difficult or fast breathing, severe chest in-drawing, convulsions or less active than usual).

asphyxia, prevention and management of hypothermia and management of pre-term and low birth weight neonates the analysis was done separately for those that delivered at home and for those that had an institutional delivery.

## Analysis

First the characteristics of women who had a birth in the 3–15 months prior to the baseline and follow-up surveys were assessed by looking at their age, education, marital status, religion, and socio-economic status. To describe the socio-economic status of the households, asset ownership, access to utilities, and household characteristics were aggregated into a single wealth index score using principal component analysis [27]. The aggregated scores were grouped into five wealth quintiles, where quintile 1 represented the poorest fifth of households and quintile 5 represented the least poor fifth. Categorical variables were summarized using percentages and continuous variables using means and confidence intervals. To estimate the changes in the coverage of the nine CBNC programme services between baseline and follow-up surveys, categorical and continuous variables were compared between the two time points using cluster adjusted F-tests and t-tests, respectively. We also assessed the changes in the coverage of indicators across the CBNC programme after controlling for the differences in the characteristics of women between baseline and follow-up surveys using a logistic regression. The Stata 13 (Stata Corporation, College Station, Texas) svy commands were used to adjust for clustering at the primary health care unit level.

## Ethics approval and consent to participate

The study was approved by the Institutional Review Boards of the London School of Hygiene and Tropical Medicine (Ethics Ref 6088, Amend No. A411) and the Ethiopian Science and Technology Ministry (Ref. No. 3.10/345/05). All respondents provided informed, voluntary written consent prior to participating in this study. Women under the age of 18 provided assent to participate in the study.

## Results

Household clusters enrolled into the study are shown in Fig 5. Of the planned 10,300 households this study included 10,224 households at baseline and 10,270 households at follow-up. In these households, we interviewed 1,016 women aged 13–49 years at baseline and 1,057 at follow-up, who had a delivery in the 3 to 15 months prior to the survey. Table 2 shows their characteristics. Study participants were similar with respect to marital status and religion. In contrast, compared to mothers interviewed at baseline, those from the follow-up survey were younger (p = 0.04) and more educated (p = 0.02). Adjusting for these factors, however, did not alter our findings with respect to changes in indicators across the nine CBNC programme components between baseline and follow-up surveys (S3–S5 Tables).

### Early identification of pregnancy

Between baseline and follow-up surveys, the proportion of women who reported having one or more ANC visits increased by 15 percentage points (95% CI: 10,19; p <0.0001) from 68% to 83% (Table 3). Woman's Development Army leaders were expected to identify and inform HEWs about pregnant women in their networks. In this study 2% (95% CI: 1,3) of women at baseline and 1% (95% CI: 1,3) at follow-up surveys informed a Woman's Development Army leader of their pregnancy (p = 0.46).

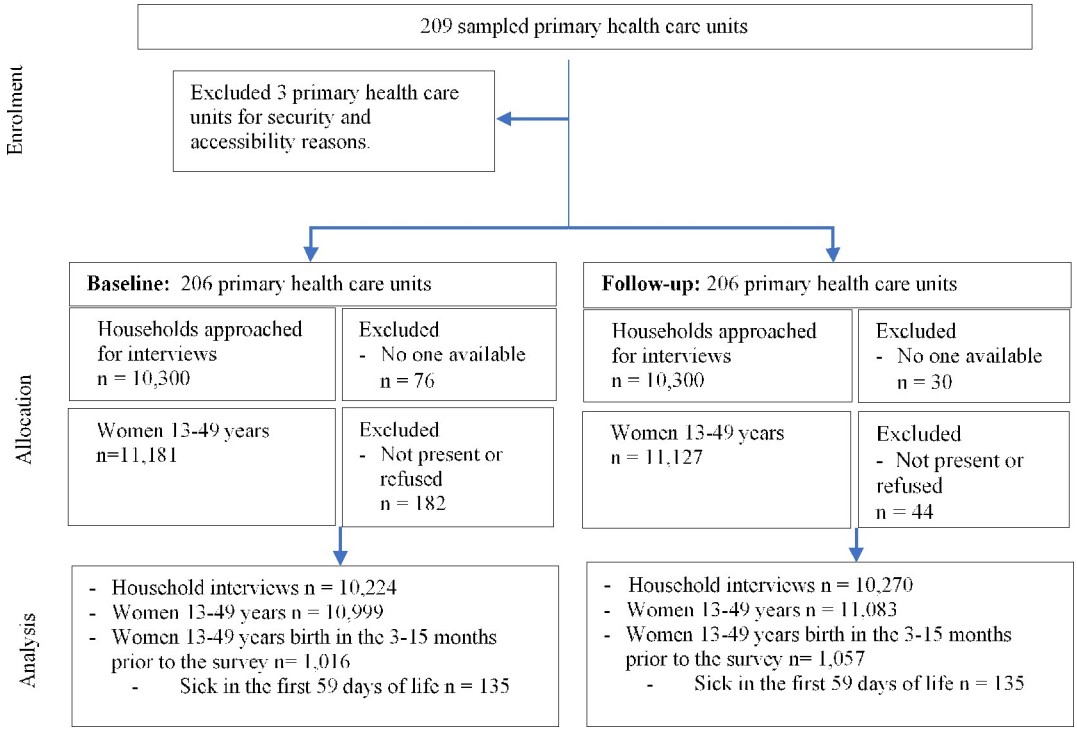

**Fig 5. Study flow diagram for baseline and follow-up surveys data collection and analysis.**

## Provision of focused ANC

Between baseline and follow-up surveys, the proportion of women who had four or more ANC visits increased by 17 percentage points (95% CI: 13,22; p <0.0001), from 36% to 53% (Table 3). Among women who had at least one ANC visit, there was an increase in the proportion who reported giving a urine sample (18 percentage point difference, 95% CI: 11,24; p<0.0001), receiving a syphilis test (8 percentage point difference, 95% CI: 3,13, p<0.0001) and receiving iron folate (9 percentage point difference, 95% CI: 3,15; p<0.02). In contrast there was a decrease in the proportion of women who said they were advised on birth preparedness (7 percentage point difference, 95% CI: -13, -1; p = 0.03), received advice on nutrition (10 percentage points (95% CI: -16,-5; p = 0.0003) and received HIV testing (19 percentage point difference, 95% CI: -24,-13; p<0.0001). The other components of focused ANC remained similar over time.

## Promotion of institutional delivery

Between baseline and follow-up surveys, the proportion of women who reported delivering in a hospital or health centre increased by 40 percentage points (95% CI: 34,44; p <0.0001), from 22% to 62% (Table 3).

## Clean and safe delivery

Among women who delivered at home, a similar proportion at baseline and follow-up surveys reported that their birth assistant washed his or her hands with soap before assisting in the delivery and that the assistant wore gloves during delivery (Table 4). For institutional deliveries there was a 15 percentage point (95% CI:-23,-7; p = 0.0002) decrease in the proportion of

**Table 2. Characteristics of mothers who had a delivery in the 3–15 months prior to the baseline (October–December 2013) and follow-up (November-December 2017) surveys.**

| Mother's characters | | Baseline survey | | Follow-up survey | | P-value |
|---|---|---|---|---|---|---|
| | | n/N | % (95% CI) | n/N | % (95% CI) | |
| Age[a] (years) | 13–19 | 79/974 | 8 (6,10) | 125/1055 | 12 (10,14) | 0.04 |
| | 20–24 | 225/974 | 23 (20,26) | 265/1055 | 25 (23,28) | |
| | 25–29 | 302/974 | 31 (28,34) | 307/1055 | 29 (26,32) | |
| | 30–34 | 213/974 | 22 (19,25) | 200/1055 | 19 (16,22) | |
| | 35–49 | 155/974 | 16 (14,19) | 158/1055 | 15 (13,17) | |
| Education[b] | No schooling | 555/971 | 57 (53,61) | 543/1057 | 51 (47,56) | 0.02 |
| | Schooling | 416/971 | 43 (39,47) | 514/1057 | 49 (44,53) | |
| Marital status[c] | Married | 901/974 | 93 (90,94) | 978/1057 | 93 (90,94) | 0.99 |
| | Not married | 73/974 | 7 (6,10) | 79/1057 | 7 (6,10) | |
| Religion[d] | Orthodox Christians | 541/974 | 55 (49,62) | 567/1057 | 54 (47,60) | 0.42 |
| | Protestant Christians | 259/974 | 27 (21,33) | 274/1057 | 26 (21,32) | |
| | Muslim | 174/974 | 18 (13,24) | 216/1057 | 20 (15,27) | |
| Socioeconomic status | Q1(poorest) | 234/1016 | 23 (20,27) | 185/1057 | 18 (15,21) | 0.10 |
| | Q2 | 177/1016 | 17 (15,20) | 204/1057 | 19 (17,22) | |
| | Q3 | 190/1016 | 19 (16,22) | 214/1057 | 20 (18,23) | |
| | Q4 | 211/1016 | 21 (18,24) | 221/1057 | 21 (18,24) | |
| | Q5(least poor) | 204/1016 | 20 (17,23) | 233/1057 | 22 (19,26) | |

Missing data on

[a] age for 42 women from the baseline survey and for 2 women from follow-up survey

[b] education status from 45 women from the baseline survey

[c] marital status for 42 women from the baseline survey; and

[d] religion for 42 women from the baseline survey.

women reporting that their birth assistant washed his or her hands with soap before assisting in the delivery, from 59% to 44% and a 6 percentage point (95% CI: -11,-2; p = 0.02) decrease from 95% to 89% in the proportion who reported their birth assistant wore gloves during delivery.

## Immediate newborn care

In both surveys, for home and institutional deliveries, one in six mothers reported use of appropriate item (clamp or new or boiled thread or string) for tying their newborn's cord (Table 4). Between baseline and follow-up surveys, the proportion of women who reported use of an appropriate item for cutting their newborn's cord (new razor blade or sterilized scissors) decreased for home deliveries by 7 percentage points (95% CI: -12,-2; p = 0.003), from 92% to 85%, with a suggestion of a similar decline for institutional deliveries. Antiseptic use for cord increased for home deliveries by 3 percentage points (95% CI: 1,5; p = 0.0001), from 1% to 4% and by 8 percentage points (95% CI: 4,11; p = 0.0003) from 3% to 11% for facility deliveries. Chlorhexidine as an antiseptic was introduced after the initiation of the CBNC programme. Among the 86 women who reported antiseptic use for cord care in the follow-up survey, 38% said the antiseptic was chlorhexidine, 14% said it was a different type of antiseptic and 43% reported not knowing the name of the antiseptic (data not shown). PNC within two days of birth for the newborn decreased for home deliveries by 6 percentage points (95% CI: -10,-3; p = 0.0001), from 8% to 2%, and also decreased for institutional delivers by 14 percentage (95% CI: -21,-7; p<0.0001), form 17% to 3%. The proportion of mothers that reported keeping

**Table 3. Antenatal care and institutional delivery for mothers who had a birth in the 3–15 months prior to the baseline (October–December 2013) and follow-up (November-December 2017) surveys.**

| | Baseline survey | | Follow-up Survey | | Percentage point difference (95% CI) | P-value |
|---|---|---|---|---|---|---|
| | n | % (95% CI) | n | % (95% CI) | | |
| **1. Early identification of pregnancy** | | | | | | |
| One or more antenatal care visit | 695/1016 | 68 (64,72) | 880/1057 | 83 (80–86) | 15 (10,19) | <0.0001 |
| Informed woman development army leader of pregnancy | 18/1016 | 2 (1,3) | 14/1057 | 1 (1–2) | -1 (-2,1) | 0.45 |
| **2. Provision of focused antenatal care** | | | | | | |
| 4 or more antenatal care visits | 362/1016 | 36 (32,39) | 560/1057 | 53 (49–57) | 17 (13,22) | <0.0001 |
| *Among those having any antenatal care visit* | | | | | | |
| First antenatal care visit at a health centre | 389/694[a] | 56 (50,62) | 535/880 | 61 (56–65) | 5 (-1.11) | 0.12 |
| Advised on birth preparedness plan | 415/694[a] | 60 (55,64) | 468/880 | 53 (49–57) | -7 (-13, -1) | 0.03 |
| Informed about pregnancy danger signs | 328/692 [b] | 47 (43,52) | 387/880 | 44 (40–48) | -3 (-10,3) | 0.25 |
| Informed about breastfeeding | 388/695 | 56 (51,61) | 507/880 | 58 (53–62) | 2 (-5,8) | 0.58 |
| Received information on nutrition | 486/693 [c] | 70 (66,74) | 524/880 | 60 (55–64) | -10 (-16,-5) | 0.0003 |
| Weight measured | 560/695 | 81 (76,84) | 709/880 | 81(77,84) | 0 (-5,5) | 0.99 |
| Height measured | 270/694[a] | 39 (34,44) | 388/880 | 44 (40,48) | 5 (-1,11) | 0.09 |
| Blood pressure measured | 509/695 | 73 (69,77) | 613/880 | 70 (66,73) | -3 (-9,1) | 0.16 |
| Gave urine sample for a test | 293/694[a] | 42 (37,47) | 526/880 | 60 (56,64) | 18 (11,24) | <0.0001 |
| Gave blood sample for syphilis test | 121/667[d] | 18 (15,22) | 232/880 | 26 (23,30) | 8 (3,13) | <0.0001 |
| Received iron folate tablets or iron syrup | 431/694[a] | 62 (57,67) | 628/880 | 71 (68,75) | 9 (3,15) | 0.002 |
| Received HIV testing | 508/694[a] | 73 (69,77) | 477/880 | 54(50,58) | -19 (-24,-13) | <0.0001 |
| **3. Promotion of institutional delivery** | | | | | | |
| Hospital or health centre delivery | 228/1015[a] | 22 (19,26) | 654/1055[c] | 62 (57,66) | 40 (35,44) | <0.0001 |

Missing data for

[a] 1 individual

[b] 3 individuals

[c] 2 individuals

[d] 28 individuals.

their newborn exclusively at home for several weeks decreased by 5 percentage points (95% CI:-9,-2; p = 0.003) for all deliveries, from 90% at baseline to 85% at follow-up survey, while the mean number of days kept at home remained constant at one month (data not shown).

## Recognition and management of asphyxia

Compared to baseline, a smaller proportion of newborns in the follow-up survey were reported to have a breathing problem at birth, irrespective of place of delivery (Table 4). Among all newborns with a breathing problem, there were no statistically significant changes in the proportion reported to have been resuscitated between baseline and follow-up surveys.

## Prevention and management of hypothermia

Although skin-to-skin contact between mother and newborn was uncommon, it increased between baseline to follow-up surveys by 11 percentage points (95% CI: 6,16; p<0.0001) for home deliveries and by 9 percentage points (95% CI: 0–17; p = 0.05) for institutional deliveries (Table 4). The proportion that reported that their newborn's bathing was delayed increased by 14 percentage points (95% CI: 6,22; p = 0. 0004) for home deliveries, from 38% to 52%, while for institutional deliveries there was no statistically significant increase.

**Table 4. Clean and safe delivery, immediate newborn care and complications management for mothers who had a birth in the 3–15 months prior to the baseline (October–December 2013) and follow-up (November-December 2017) surveys.**

| | Home deliveries | | | | percentage point difference (95% CI) | P-value | Institutional deliveries | | | | percentage difference (95% CI | P-value |
|---|---|---|---|---|---|---|---|---|---|---|---|---|
| | Baseline survey | | Follow-up survey | | | | Baseline survey | | Follow-up Survey | | | |
| | n/N | % (95% CI) | n/N | % (95% CI) | | | n/N | % (95% CI) | n/N | % (95% CI) | | |
| **4. Clean and safe Delivery** | | | | | | | | | | | | |
| Birth assistant washed hands with soap [a] | 457/ 767[a] | 60 (57,65) | 197/ 358[b] | 55 (49,61) | -6 (-12,1) | 0.09 | 135/ 228 | 59 (52,66) | 287/ 654 | 44 (39,49) | -15 (-23, -7) | 0.0002 |
| Birth assistant wore gloves [b] | 126/ 763[c] | 17 (14,21) | 72/ 358[b] | 20 (16,25) | 3 (-3,9) | 0.26 | 217/ 228 | 95 (90,98) | 579/ 654 | 89 (86,91) | -6 (-11,-2) | 0.02 |
| Delivery took place on a clean surface[c] | 688/ 768[d] | 90 (87,93) | 254/ 358[b] | 71 (66,76) | -19 (-25,-14) | <0.0001 | 216/ 228 | 95 (90,97) | 608/ 654 | 93 (91,95) | -2 (-6,3) | 0.45 |
| **5. Immediate newborn care** | | | | | | | | | | | | |
| Clamp or new or boiled thread or string used to tie cord | 461/ 781[e] | 59 (54,64) | 256/ 399[f] | 64 (58,70) | 5 (-2,12) | 0.17 | 137/ 223[g] | 61 (54,69) | 406/ 650[h] | 62 (58,67) | 1 (-8,10) | 0.82 |
| New razor blade or sterilized scissors used to cut cord | 719/ 782[g] | 92 (89,93) | 340/ 399[f] | 85 (80,89) | -7 (-12,-2) | 0.003 | 158/ 223[g] | 71 (63,77) | 421/ 650[h] | 65 (60,69) | -6 (-14,2) | 0.13 |
| Antiseptic used on the cord | 4/772[i] | 1 (<1,2) | 14/ 399[f] | 4 (2,6) | 3 (1,5) | 0.0001 | 7/218[j] | 3 (2,6) | 72/ 650[h] | 11 (9,14) | 8 (4,11) | 0.0003 |
| Newborn postnatal check in first 2 days | 61/ 782[g] | 8 (5,11) | 7/399[f] | 2 (1,4) | -6 (-10,-3) | 0.0001 | 39/ 224[h] | 17 (11,26) | 20/ 650[h] | 3 (2,5) | -14 (-21,-7) | <0.0001 |
| **6. Recognition and management of asphyxia** | | | | | | | | | | | | |
| Newborn with difficulty crying/ breathing | 70/ 772[i] | 9 (7,12) | 12/ 399[f] | 3 (2,5) | -6 (-9,-3) | 0.0001 | 36/ 219[k] | 16 (12,23) | 51/ 650[h] | 8 (6,10) | -8 (-14,-3) | 0.0008 |
| Among them, resuscitated newborns | 5/67[l] | 7 (3,17) | 2/12 | 17 (4,49) | 10 (-13,32) | 0.31 | 17/ 35[m] | 49 (32,66) | 18/51 | 35 (23,50) | -14 (-34,8) | 0.21 |
| **7. Prevention and management of hypothermia** | | | | | | | | | | | | |
| Newborn placed on mother's belly or chest immediate after birth | 107/ 766[n] | 14 (11,18) | 100/ 399[f] | 25 (20,30) | 11 (6,16) | <0.0001 | 62/ 218[j] | 28 (22,36) | 242/ 650[h] | 37 (33,42) | 9 (0–17) | 0.05 |
| Bathing of newborn delayed for 24 hrs | 290/ 772[i] | 38 (33–43) | 207/ 399[f] | 52 (46–58) | 14 (6,22) | 0.0004 | 167/ 218[j] | 77 (70–82) | 519/ 650[h] | 80 (76–83) | 3 (-4,10) | 0.34 |
| **8. Management of pre-term and low birth weight neonates** | | | | | | | | | | | | |
| Newborn weighed at birth | 36/ 778[k] | 5 (3–7) | 26/ 399[f] | 7 (4–10) | 2 (-1,5) | 0.13 | 138/ 222[e] | 62 (55–69) | 395/ 650[h] | 61 (56–65) | -1 (-9,7) | 0.73 |

Missing data for

[a] 20 individuals

[b] 43 individuals

[c] for 24 individuals

[d] 19 individuals

[e] 6 individuals

[f] 2 individuals

[g] 5 individuals

[h] 4 individuals

[i] 15 individuals

[j] 10 individuals

[k] 9 individuals

[l] 3 individuals

[m] 1 individual; and

[n] 21 individuals.

**Table 5. Treatment of sick young infants reported by mothers who had a birth in the 3–15 months prior to the baseline (October–December 2013) and follow-up (November-December 2017) surveys.**

| Management of possible serious bacterial infection at community level | Baseline survey | | Follow-up Survey | | % point difference (95% CI) | P-value |
|---|---|---|---|---|---|---|
| | n/N | % (95% CI) | n/N | % (95% CI) | | |
| Children sick in the first 59 days of life | 135/951[a] | 14 (12,17) | 135/1051[b] | 13 (12,15) | -1 (-5,2) | 0.41 |
| Symptoms of possible serious bacterial infection[c] | 98/951[a] | 10 (8,13) | 103/1051[b] | 10 (8,12) | -1 (-4,3) | 0.65 |
| *Among young infants with possible serious bacterial infection* | | | | | | |
| Amoxicillin for 7 days | 19/98 | 19 (12,30) | 71/103 | 69 (59,77) | 50 (37,62) | <0.0001 |
| Gentamicin for 7 days | 7/98 | 7 (3,16) | 23/103 | 22 (15,32) | 15 (5,25) | 0.007 |
| Amoxicillin and gentamicin and for 7 days | 3/98 | 3 (1,9) | 15/103 | 15 (9,23) | 12 (4,19) | 0.006 |

Missing data for

[a] 65 individuals; and

[b] 6 individuals.

[c] Signs of possible serious bacterial infection: unusually hot or unusually cold, reduced feeding, difficult or fast breathing, serious chest in-drawing, convulsions or less active than usual (lethargy).

## Management of pre-term and low birth weight neonates

In both baseline and follow-up surveys, one out of ten newborns were reported to have been weighed at birth for home deliveries and 6 out of ten newborns were reported to have been weighed for facility deliveries (Table 4).

## Management of possible serious bacterial infection at community level

A similar number of children were reported to have an illness in the first 59 days of life at baseline (14%, 95% CI: 12,17) and follow-up (13%, 95% CI: 12,15) surveys, and one in ten reportedly had signs of possible serious bacterial infection (Table 5). Among cases of possible serious bacterial infection, amoxicillin for seven days increased by 50 percentage points (95% CI: 37,62; p<0.0001) from 19% at baseline to 69% at follow-up survey, and gentamicin injection for seven days increased by 15 percentage points (95% CI: 5,25; p = 0.0007) from 7% to 22%. Concurrent use of both antibiotics increased overtime by 12 percentage points (95% CI: 4,19; p = 0.006), from 3% to 15%.

## Discussion

This study found that, when comparing service coverage before and after the CBNC programme implementation, marked improvements were seen for early identification of pregnancy, focused ANC, health facility delivery, prevention and management of hypothermia for home deliveries and antibiotic provision for the treatment of possible serious bacterial infection. Very few newborns at the follow-up survey were reported to have used antiseptic use for cord care, even though there had been some improvement over time. We found no evidence of improvement over time for the remaining programme components of CBNC, namely safe and clean delivery, recognition and management of asphyxia, and management of pre-term and low birth weight neonates.

At baseline and follow-up surveys, mothers were asked about their pregnancy, delivery, newborn and sick young infant care for the 3–15 months prior to the date of the surveys. Births further back from the date of the survey might have been recalled differently from more recent births, however this is unlikely to affect our findings as the same approach was used at baseline and follow-up surveys. Although the tools were pre-tested, some questions relied on mothers' understanding of technical concepts (e.g. presence of a breathing problem for a newborn).

This might have underestimated or overestimated coverage of some indicators. However, as the same tools were used at baseline and follow-up surveys it does not affect the comparisons made over time. Some indicators such as the management of new-borns with low birth weight were partially measured as most mothers did not know the weight of their newborns at birth, which made it challenging to identify those that had a low birth weight. Data from facility registers also did not provide a solution as information on the management of low birth weight and asphyxiated newborns were rarely or poorly recorded. The districts included in this study were representative of their zones. However, zones were not randomly selected and are thus not a representative sample of the regions. As such generalization of the study findings to the whole country should be made with caution. Although the baseline and follow-up survey samples differed with respect to mothers' age distribution and their educational level, adjusting for these potential confounders did not alter the findings. However, some of the differences in service coverage might be due to unmeasured confounders, that could have been addressed with the use of a concurrent comparison group that was not exposed to the CBNC programme. A large number of significance tests were carried out and while we made no formal adjustment for multiple comparisons, we recognise the possibility of an increased number of false positives and interpret our findings with due caution.

Despite the increased report of women having at least one ANC in this study, very few women said that they had informed the Woman's Development Army leaders about their pregnancy, indicating that the role of volunteers in supporting early identification of pregnancy was not actualized. Women having four or more ANC visits increased and, although lower in proportion, the Ethiopian Demographic and Health surveys show a similar trend. Between 2011 and 2016 four or more ANC visits for the most recent birth in the five years preceding the Demographic and Health Survey increased from 19% to 33% [4, 28]. However, contact with health service providers along the continuum of care might not serve as a good proxy indicator for the coverage of life saving interventions [29]. Although in this study, as seen in other local and global studies, the proportion of women who received iron and folate and those that gave a blood sample for syphilis testing increased over time, the overall content of the ANC visits showed that most women were still not getting the care they needed [30, 31]. In addition to increasing the number of ANC contacts, improving the quality of care is necessary to improve the coverage of life saving interventions [32].

The proportion of women reporting an institutional delivery tripled between baseline and follow-up surveys. Similar trends have been shown in other studies including the Ethiopian Demographic and Health Survey, which showed an increase from 10% in 2011 to 26% in 2016 [4, 28, 33]. This increase in institutional delivery has been catalysed in part by the by the saturation of health messages that promote facility delivery, improved accessibility to health facilities and prohibiting use of traditional birth attendants. The WDA strategy, which the CBNC programme relied on, has also been credited for improving maternal health practices. One study has shown active WDA networks was associated with better coverage of facility deliveries [18].

Only one in 10 newborns were reported to have received antiseptic during the follow-up survey. The health facility assessment done as part of the follow-up survey also showed that only 15% of health posts and 62% of health centres had chlorhexidine on the day of the survey indicating the need to strengthen the capacity of the supply chain system for CBNC programme related supplies [34]. However, our findings on the reported use of antiseptic for cord care and other care provided to mothers and newborns around the time of delivery, such as birth assistant washing hands or wearing gloves and weighing of the newborn need to be interpreted with caution. A study done in Nigeria found that while mothers' exit-interview report had high accuracy for 9 out of 20 child birth care indicators, follow-up interviews at 9 to 22 months after child birth showed that only placing newborn skin-to-skin was accurately recalled [35].

Despite the aim of the CBNC programme to increase early contact with newborns, this study found an alarmingly low level of PNC visits in the first 48 hours after birth at baseline, which decreased to 5% at the follow-up survey. This is particularly striking given the increase in ANC visits and facility delivery. While the level of PNC coverage seen in this study was lower than the 15% reported in the 2016 Ethiopian Demographic and Health Survey, other studies report similar findings [28, 33, 36]. Studies have shown that the lack of a functioning mechanism for alerting HEWs to a birth, home deliveries, accessibility and distance between homes and health posts, HEWs workload and motivation hinder PNC service provision [37–39]. Furthermore, in this study 9 out of 10 mothers kept their newborn at home for the first few weeks of life. Such cultural postpartum restrictions are additional barriers to seeking PNC services [40, 41]. The postnatal period poses a great risk for neonatal survival, and one which is key for providing lifesaving interventions [42]. Improving the support provided to HEWs from health centre staff and Woman's Development Army leaders, including notifying HEWs of facility and home deliveries, could lead to better home based coverage of PNC [43, 44].

Maternal reports of antibiotic treatment for possible serious bacterial infections in young infants may have limitations if parallels are drawn with studies showing that household surveys may not provide a reliable estimate of treatment for possible pneumonia due to the likelihood of including non-cases in the denominator when calculating the antibiotic treatment rate [45, 46]. Acknowledging this limitation, this study showed an increase in the reported use of antibiotics for 7 days for young infants with possible serious bacterial infection, particularly amoxicillin for seven days. Although the concurrent use of amoxicillin and gentamycin for seven days increased from 3% at baseline to 15% at follow-up, most of the treatments were not according to national guidelines which specify provision of both antibiotics. This is indicative of the quality of care provided by HEWs. A study that assessed the quality of care provided by HEWs showed that they misdiagnosed 70% of every severe bacterial infections in young infants [47]. Similar findings were also observed for their management of older children 2–59 months of age [48–50]. Furthermore, the low use of antibiotics could be due to stockout of drugs, particularly gentamicin. At the time of the follow-up survey, while 79% of health posts had amoxicillin, only 35% had gentamicin [34]. To expedite the service initiation for the management of possible serious bacterial infection at health posts, implementing partners purchased and delivered gentamicin to districts in parallel to the existing system, as gentamicin 20 mg/2 ml was not part of the country's essential drugs list. In order to ensure the sustainability of this service, gentamicin must be integrated into the supply chain management system [51]. Furthermore, HEWs may not be available for seven consecutive days to provide gentamicin injections. The World Health Organization's 2015 recommendation to provide oral amoxicillin to sick young infants with only fast breathing and to treat those with other symptoms of possible serious bacterial infection with two days of gentamicin and 7-days oral amoxicillin can provide a solution to these problems [52]. In 2019 the simplified antibiotic regimen was adopted into the CBNC programme guidelines. The CBNC programme initiative stemmed in part from the Community-Based Interventions for Newborns in Ethiopia trial [25]. Findings from the trial, which were published after the launch of the CBNC programme, showed weak evidence of the impact of HEWs identifying and managing newborns with possible serious bacterial infection on neonatal mortality. This implies that perhaps the programme did not benefit from emerging evidence on the health system constraints as well as the importance of active case seeking by volunteers and the need for a comprehensive plan to improve care seeking behaviour.

## Conclusion

The CBNC programme is an ambitious and innovative attempt to address newborn deaths in a challenging context. The before-and-after comparison of the CBNC programme service

coverage in four regions of Ethiopia showed improvements in use of ANC and institutional delivery. Some components of care provided to mothers and newborns around the time of delivery also improved, while PNC remained alarmingly low. HEWs urgently need additional support both from Woman Development Army leaders and health centre staff, allowing them to a have a more proactive engagement in postnatal care visits to women at home. The CBNC programme is a progressive step towards preventing deaths from possible serious bacterial infection but treatment remains inadequate. A simplified antibiotic regimen is likely to improve access to, and completion of antibiotics.

## Supporting information

**S1 File.**
(PDF)

**S2 File.**
(PDF)

**S3 File.**
(PDF)

**S4 File.**
(PDF)

**S1 Table. Description of the Community-Based Newborn Care programme intervention.**
(PDF)

**S2 Table. Description of the Community-Based Newborn Care programme training components.**
(PDF)

**S3 Table. Antenatal care and institutional delivery for mothers who had a birth in the 3–15 months prior to the baseline (October–December 2013) and follow-up (November-December 2017) surveys.**
(PDF)

**S4 Table. Clean and safe delivery, immediate newborn care and complications management for mothers who had a birth in the 3–15 months prior to the baseline (October–December 2013) and follow-up (November-December 2017) surveys.**
(PDF)

**S5 Table. Treatment of sick young infants reported by mothers who had a birth in the 3–15 months prior to the baseline (October–December 2013) and follow-up (November-December 2017) surveys.**
(PDF)

## Acknowledgments

The authors would like to acknowledge the assistance of the Ethiopian Federal Ministry of Health, the CBNC programme steering committee as well as UNICEF, Save the Children, and Last 10 Kilometres/John Snow Inc., and Integrated Family Health Programme. The authors would also like to thank Tsegahun Tessema from JaRco Consulting for his unwavering support during the CBNC programme evaluation. A special thanks to Shimiljash Braha and Mahader Tamene for reviewing documents to map the CBNC programme interventions. Lastly, we

would like to thank all those who were involved in the data collection and study participants who agreed to generously give their time to participate in study.

## Author Contributions

**Conceptualization:** Della Berhanu, Elizabeth Allen, Joanna Schellenberg, Bilal Iqbal Avan.

**Data curation:** Della Berhanu, Elizabeth Allen, Emma Beaumont, Keith Tomlin, Nolawi Taddesse, Girmaye Dinsa, Yirgalem Mekonnen, Hanna Hailu, Manuela Balliet, Neil Lensink, Joanna Schellenberg, Bilal Iqbal Avan.

**Formal analysis:** Della Berhanu, Emma Beaumont.

**Funding acquisition:** Joanna Schellenberg, Bilal Iqbal Avan.

**Investigation:** Della Berhanu, Joanna Schellenberg, Bilal Iqbal Avan.

**Methodology:** Della Berhanu, Elizabeth Allen, Joanna Schellenberg, Bilal Iqbal Avan.

**Project administration:** Della Berhanu, Joanna Schellenberg, Bilal Iqbal Avan.

**Resources:** Joanna Schellenberg.

**Supervision:** Della Berhanu, Nolawi Taddesse, Girmaye Dinsa, Yirgalem Mekonnen, Neil Lensink, Bilal Iqbal Avan.

**Writing – original draft:** Della Berhanu.

**Writing – review & editing:** Elizabeth Allen, Emma Beaumont, Keith Tomlin, Nolawi Taddesse, Girmaye Dinsa, Yirgalem Mekonnen, Hanna Hailu, Manuela Balliet, Neil Lensink, Joanna Schellenberg, Bilal Iqbal Avan.

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
