## [Decision Letter · Decision Letter 0]

29 Oct 2020

PONE-D-20-24389

Community-Based Newborn Care in Ethiopia: effects of a national programme on antenatal, intrapartum, and newborn care in 104 districts.

PLOS ONE

Dear Dr. Berhanu,

Thank you for submitting your manuscript to PLOS ONE. After careful consideration, we feel that it has merit but does not fully meet PLOS ONE’s publication criteria as it currently stands. Therefore, we invite you to submit a revised version of the manuscript that addresses the points raised during the review process.

This is an important issue for universal health coverage, well planned analysis and structure written manuscript. But the methodology is weak/limited and many variables need to be address adequately (eg., quality of care). Therefore,  suggestion to revise the draft in order to desire the intention of the analyses and actual ability given the data offers. 

We look forward to receiving your revised manuscript.

Kind regards,

Mahfuzar Rahman, MD, PhD

Academic Editor

PLOS ONE

Journal Requirements:

2.  Thank you for stating in your ethics statement "For women under the age of 18 assent was obtained from parent or guardian." Generally for people under the age of 18, parents or guardians give informed consent and minors provide assent. Please clarify if this is what is meant."

3. Please include additional information regarding the survey or questionnaire used in the study and ensure that you have provided sufficient details that others could replicate the analyses. For instance, if you developed a questionnaire as part of this study and it is not under a copyright more restrictive than CC-BY, please include a copy, in both the original language and English, as Supporting Information."

6. We note that [Figure 2] in your submission contains a map image which may be copyrighted. All PLOS content is published under the Creative Commons Attribution License (CC BY 4.0), which means that the manuscript, images, and Supporting Information files will be freely available online, and any third party is permitted to access, download, copy, distribute, and use these materials in any way, even commercially, with proper attribution. For these reasons, we cannot publish previously copyrighted maps or satellite images created using proprietary data, such as Google software (Google Maps, Street View, and Earth). For more information, see our copyright guidelines: http://journals.plos.org/plosone/s/licenses-and-copyright.

1.     You may seek permission from the original copyright holder of Figure(s) [2] to publish the content specifically under the CC BY 4.0 license.  

Reviewers' comments:

Reviewer's Responses to Questions

**Comments to the Author**

1. Is the manuscript technically sound, and do the data support the conclusions?

Reviewer #1: Yes

Reviewer #2: No

2. Has the statistical analysis been performed appropriately and rigorously? 

Reviewer #1: No

Reviewer #2: Yes

3. Have the authors made all data underlying the findings in their manuscript fully available?

Reviewer #1: No

Reviewer #2: Yes

4. Is the manuscript presented in an intelligible fashion and written in standard English?

Reviewer #1: Yes

Reviewer #2: Yes

5. Review Comments to the Author

Reviewer #1: I congratulate the authors for preparing a manuscript on an important topic, as community-based programs has become an increasingly popular health sector development strategy to expand life-saving health services in resource-constrained

settings. The manuscript adequately described the methodology used and the findings are explained well. My first major comment is as follows: given the changes in respondents' background characteristics (viz., age, educational attainment), the authors should consider reporting the results from regression analyses. Though the authors noted that the results remain unchanged, control for multiple key variables that might otherwise bias the results would be useful.

My second major comment would be a discussion on quality of care. An earlier paper by Miller and colleagues (2014) found that only 34% of children with severe illness were correctly managed under the iCCM program in Ethiopia (doi:10.4269/ajtmh.13-0751). Your study results indicate that the lack of improvement in some of the key CBNC components would be attributable to quality of care issue as well (in addition to provider training, sustained supervision, and availability of essential commodities).

The manuscript needs a review to fix some minor typos (e.g., lines 77 and 90 of page 4).

Reviewer #2: This study attempts to measure the effects of a community based newborn care program in Ethiopia on the coverage of key maternal and newborn interventions. It relies on high quality primary data and investigates some elements of the care continuum that are often unmeasured such as recognition of asphyxia.

However, because of the changes in the rollout to the intervention, the data is wholly inadequate to answering the question that they have posed about the effect of the intervention due to the lack of an appropriate comparison group. I am sympathetic to the change in rollout that the researchers had no control over and recognize that they were transparent about the limitations of the study. For example, in the discussion they mention a litany of other causes that may have resulted in the increased institutional delivery including improved accessibility and prohibiting the use of traditional birth attendants, both of which were unrelated to the CBNC intervention. They also note that the woman development army leaders were not notified, indicating that the program did not work as intended. However, much of their framing was still problematic including the title (e.g. “effects of”) and their research question (e.g. “aimed to assess the extent to which the CBNC programme increased the coverage of services”), both of which use causal language.

I see two potential solutions to this mismatch between intent and actual ability given the data. First, the study could be reframed to discuss how elements of CBNC coverage have changed over time. Less emphasis would then be placed on the CBNC interventions and rollout, and the estimates could be compared with DHS. This would make use of the data collected. The second option, if the authors were interested in answering their current research question, would be to use an alternate dataset such as the DHS, which has data from the control and intervention districts over the relevant time period on many of the indicators in this study. This would make the conclusions about the efficacy of the CBNC program much stronger.

Other major comments:

- Many of the outcome indicators selected seemed to only partially measure the program components. For example, for management of pre-term and low birth weight neonates, there was just a single measure on whether the newborn was weighed, which doesn’t indicate management at all. Were any other measures on KMC available, for example? Additionally, there needs to be more explanation about why some of the measures on management of pre-term and low birth weight neonates and management of asphyxia were not collected at the health facility, which seems more appropriate than women’s self-report.

- I didn’t understand the logic of how the CBNC program, which was primarily community based through the HEWs and women’s development army, was meant to affect actions occurring within health facilities on immediate newborn care, recognition and management of asphyxia and management of preterm and low birth weight neonates. The statement “Of the CBNC programme components, seven of them were part of the existing services,” was also perplexing: what was the purpose of repackaging existing services in a new program name? Would a retooled evaluation only estimate the effect of the two new components?

Minor comments:

- Proofread for typos throughout (i.e. pg 4 line 90, pg 9 line 193)

- Figure 2 requires a legend

6. PLOS authors have the option to publish the peer review history of their article (what does this mean?). If published, this will include your full peer review and any attached files.

Reviewer #1: No

Reviewer #2: No

---

## [Author Response · Author response to Decision Letter 0]

8 Apr 2021

Reviewers' comments:

Reviewer's Responses to Questions

Comments to the Author

5. Review Comments to the Author

Reviewer #1: I congratulate the authors for preparing a manuscript on an important topic, as community-based programs has become an increasingly popular health sector development strategy to expand life-saving health services in resource-constrained

settings. The manuscript adequately described the methodology used and the findings are explained well. 

Response: Thank you very much. 

My first major comment is as follows: given the changes in respondents' background characteristics (viz., age, educational attainment), the authors should consider reporting the results from regression analyses. Though the authors noted that the results remain unchanged, control for multiple key variables that might otherwise bias the results would be useful.

Response: Thank you for this comment. As mentioned, we did consider this option and also ran some logistic regressions to see if there was any evidence of confounding, in terms of a change in the ORs after adjustment. Based on the reviewers comment we have re-checked this exhaustively for all indicators and produced a supplementary table which could be included at the discretion of the editor. The results suggest remarkably little evidence of confounding: of 45 indicators, just three had an adjusted odds ratio more than 15% different from the unadjusted odds ratio. Of these three, two (informed WDA leader and resuscitated newborns who had a breathing problem for home deliveries) showed no evidence of an association with the CBNC programme, and this conclusion was unchanged by adjustment. The third (amoxicillin for 7 days) showed strong evidence of an association with the CBNC programme (OR 9, 95CI 5,19) which remained after adjustment (OR 12, 95% CI 5,27).

My second major comment would be a discussion on quality of care. An earlier paper by Miller and colleagues (2014) found that only 34% of children with severe illness were correctly managed under the iCCM program in Ethiopia (doi:10.4269/ajtmh.13-0751). Your study results indicate that the lack of improvement in some of the key CBNC components would be attributable to quality of care issue as well (in addition to provider training, sustained supervision, and availability of essential commodities).

Response: Thank you for indicating that the discussion on quality of care was missing. We have now (on line 589) added this discussion along with 4 references addressing that HEWs misdiagnosed 70% of young infants with possible severe bacterial infection (Berhanu et al), that they had poor diagnostic capacity when managing older children (Getachew et al and Miller et al), and that they also, to a large extent, did not follow treatment guidelines when managing sick children (Daka et al). 

The manuscript needs a review to fix some minor typos (e.g., lines 77 and 90 of page 4).

Response: Thank you. We have fixed this by removing CBNC from line 84 and fixed the “Th” to “Th” on line 97. 

Reviewer #2: This study attempts to measure the effects of a community based newborn care program in Ethiopia on the coverage of key maternal and newborn interventions. It relies on high quality primary data and investigates some elements of the care continuum that are often unmeasured such as recognition of asphyxia.

However, because of the changes in the rollout to the intervention, the data is wholly inadequate to answering the question that they have posed about the effect of the intervention due to the lack of an appropriate comparison group. I am sympathetic to the change in rollout that the researchers had no control over and recognize that they were transparent about the limitations of the study. For example, in the discussion they mention a litany of other causes that may have resulted in the increased institutional delivery including improved accessibility and prohibiting the use of traditional birth attendants, both of which were unrelated to the CBNC intervention. They also note that the woman development army leaders were not notified, indicating that the program did not work as intended. However, much of their framing was still problematic including the title (e.g. “effects of”) and their research question (e.g. “aimed to assess the extent to which the CBNC programme increased the coverage of services”), both of which use causal language. I see two potential solutions to this mismatch between intent and actual ability given the data. First, the study could be reframed to discuss how elements of CBNC coverage have changed over time. Less emphasis would then be placed on the CBNC interventions and rollout, and the estimates could be compared with DHS. This would make use of the data collected. The second option, if the authors were interested in answering their current research question, would be to use an alternate dataset such as the DHS, which has data from the control and intervention districts over the relevant time period on many of the indicators in this study. This would make the conclusions about the efficacy of the CBNC program much stronger.

Response: We appreciate the importance of this comment. We have changed the language (“effects of” and “aimed to assess the extent to which the CBNC programme increased the coverage of services”), which infer causal relationship between CBNC programme and different outcomes. Although we found the second suggestion interesting, the DHS data would not allow us to answer our original question as CBNC was rolled out throughout the agrarian regions of the country by the time the endline survey was conducted. As a result, we have taken the first suggestion and reframed the paper to discuss how elements of the CBNC coverage have changed over time. In the discussion section, we have compared the estimates with indicators available in the DHS, mainly for antenatal care (line 531), institutional delivery (line 543) and postnatal care (line 568 ). Where possible, we have also clarified how some of the CBNC strategies might have been linked with the observed increases. For example, we have added text and a reference (Damtew et al) in the discussion section on line 547, showing how the WDA strategy, which the CBNC programme attempted to strengthen, has been shown to increase maternal health practices, including institutional deliveries. 

Other major comments:

- Many of the outcome indicators selected seemed to only partially measure the program components. For example, for management of pre-term and low birth weight neonates, there was just a single measure on whether the newborn was weighed, which doesn’t indicate management at all. Were any other measures on KMC available, for example? 

Response: We agree that some of the outcomes have a single measure that partially captures the necessary information. We found measuring some of the indicators like KMC challenging. For example among the women in this study that had their newborn weighed (17% at baseline and 40% at endline), a majority (45% at baseline and 59% at endline) did not know what their weight was, nor did they have any written record of it. Hence when we restrict to those that were low birth weight and then ask about KMC, we only have data on very few newborns. As such we did not include the information in the table. However, we have indicated this in the limitations section on line 510.

Additionally, there needs to be more explanation about why some of the measures on management of pre-term and low birth weight neonates and management of asphyxia were not collected at the health facility, which seems more appropriate than women’s self-report.

Response: We agree with this comment and had in the baseline study attempted to collect this data from health center registers. However, data on these indicators were rarely or poorly recorded in facility registers. We have also added this point in the discussion section on line 513.

- I didn’t understand the logic of how the CBNC program, which was primarily community based through the HEWs and women’s development army, was meant to affect actions occurring within health facilities on immediate newborn care, recognition and management of asphyxia and management of preterm and low birth weight neonates. 

Response: we agree that this requires further clarification. The 4th (safe and clean delivery), 5th (immediate newborn care) and 6th (recognition of asphyxia and initial stimulation) components of the CBNC programme aimed to target non-facility deliveries. HEWs and WDA leaders were expected to promote deliveries in hospitals and health centers, but in instances where HEWs had to attend a home or health post-deliveries, they were expected to perform these tasks. Hence in this study we attempted show the coverage of these services by separating home and facility deliveries. We have added this information on line 184. The detailed components of the CBNC programme implementation and training components are provided in Supplementary Table 1 and 2.

The statement “Of the CBNC programme components, seven of them were part of the existing services,” was also perplexing: what was the purpose of repackaging existing services in a new program name? Would a retooled evaluation only estimate the effect of the two new components?

Response: the CBNC programme relied on the 4 Cs model (line 151): C1-early contact with pregnant mothers and newborns; C2-case identification of possible severe bacterial infection in young infants; C3-care and treatment of young infants with possible severe bacterial infection; and, C4-completion of their 7 days of antibiotic treatment. With this model the programme aimed to strengthen the health system and support HEWs to identify pregnant women, promote facility delivery (or ensure safe and clean delivery for non-facility deliveries), which were believed to be pathways to identifying sick young infants. Hence this study aimed to assessed if coverage of both the existing and new services changed after the launch of the CBNC programme. 

Minor comments:

- Proofread for typos throughout (i.e. pg 4 line 90, pg 9 line 193)

Response: thank you for catching these errors. We have fixed the “Th” to “The” on page 4 and corrected the year to 2014 on line 203.

- Figure 2 requires a legend

Response: Thank you. In the map now we have added a figure legend that describes the different shades of the regions. 

---

## [Decision Letter · Decision Letter 1]

3 May 2021

Coverage of antenatal, intrapartum, and newborn care in 104 districts of Ethiopia: a before and after study four years after the launch of the national Community-Based Newborn Care programme

PONE-D-20-24389R1

Dear Dr. Berhanu,

We’re pleased to inform you that your manuscript has been judged scientifically suitable for publication and will be formally accepted for publication once it meets all outstanding technical requirements.

Kind regards,

Mahfuzar Rahman, MD, PhD

Academic Editor

PLOS ONE

Additional Editor Comments (optional):

Reviewers' comments:

Reviewer's Responses to Questions

**Comments to the Author**

1. If the authors have adequately addressed your comments raised in a previous round of review and you feel that this manuscript is now acceptable for publication, you may indicate that here to bypass the “Comments to the Author” section, enter your conflict of interest statement in the “Confidential to Editor” section, and submit your "Accept" recommendation.

Reviewer #2: All comments have been addressed

2. Is the manuscript technically sound, and do the data support the conclusions?

Reviewer #2: Yes

3. Has the statistical analysis been performed appropriately and rigorously? 

Reviewer #2: Yes

4. Have the authors made all data underlying the findings in their manuscript fully available?

Reviewer #2: Yes

5. Is the manuscript presented in an intelligible fashion and written in standard English?

Reviewer #2: Yes

6. Review Comments to the Author

Reviewer #2: Thank you for your careful revision and responses. All of my previous comments have been addressed, and the manuscript is substantially improved.

7. PLOS authors have the option to publish the peer review history of their article (what does this mean?). If published, this will include your full peer review and any attached files.

Reviewer #2: **Yes: **Anna Gage

---

## [Editor Report · Acceptance letter]

21 Jul 2021

PONE-D-20-24389R1 

Coverage of antenatal, intrapartum, and newborn care in 104 districts of Ethiopia: a before and after study four years after the launch of the national Community-Based Newborn Care programme 

Dear Dr. Berhanu:

I'm pleased to inform you that your manuscript has been deemed suitable for publication in PLOS ONE. Congratulations! Your manuscript is now with our production department. 

Kind regards, 

on behalf of

Dr. Mahfuzar Rahman 

Academic Editor

PLOS ONE